

# Improving discrimination between clouds and optically thick aerosol plumes in geostationary satellite data

Daniel Robbins[1,2], Caroline Poulsen[1,3], Steven Siems[1,2,4], and Simon Proud[5,6]

[1]School of Earth, Atmosphere and Environment, Monash University, Melbourne, VIC 3800, Australia
[2]ARC Centre of Excellence for Climate Extremes, Monash University, Melbourne, VIC 3800, Australia
[3]Bureau of Meteorology, Melbourne, VIC, 3001, Australia
[4]ARC SRI Securing Antarctica's Environmental Future, Melbourne, VIC 3800, Australia
[5]National Centre for Earth Observation, University of Oxford, Clarendon Laboratory, Parks Road, Oxford, OX1 3PU, UK
[6]STFC RAL Space and the National Centre for Earth Observation, Rutherford Appleton Laboratory, Didcot, OX11 0QX, UK

**Correspondence:** Daniel Robbins (daniel.robbins@monash.edu)

**Abstract.** Cloud masking is a key initial step in the retrieval of geophysical properties from satellite data. Despite decades of research problems still exists of over or under detection of cloud. High aerosol loadings, in particular from dust storms or fires, are often classified as cloud and vice versa. In this paper we present a cloud mask created using machine learning for the Advanced Himawari Imager (AHI) on board Himawari-8. In order to train the algorithm a parallax-corrected collocated data set

was created from AHI and Cloud-Aerosol Lidar with Orthogonal Polarization (CALIOP) lidar data. Artificial neural networks (ANNs) were trained on the collocated data to identify clouds in AHI scenes. The resulting NN cloud masks are validated and compared to cloud masks produced by the Japanese Meteorological Association (JMA) and the Bureau of Meteorology (BoM) for a number of different solar and viewing geometries, surface types and airmasses. 5 case studies covering a range of challenging scenarios for cloud masks are also presented to demonstrate the performance of the masking algorithm. The

NN mask shows a lower false positive rate (FPR) for an equivalent true positive rate (TPR) across all categories, with FPRs of 0.106 and 0.198 for the NN and JMA masks respectively and 0.314 and 0.464 for the NN and BoM masks respectively at equivalent TPR values. This indicates the NN mask accurately identifies 1.11 and 1.28 times as many non-cloud pixels, for the equivalent hit rate when compared to the JMA and BoM masks respectively. The NN mask was shown to be particularly effective in distinguishing thick aerosol plumes from cloud, most likely due to the inclusion of the 0.47μm and 0.51μm bands.

The NN cloud mask shows an improvement over current operational cloud masks in most scenario and it is suggested that improvements to current operational cloud masks could be made by including the 0.47μm and 0.51μm bands. The collocated data are made available to facilitate future research.

## 1 Introduction

Earth observing satellite instruments are critical for measuring geophysical parameters for weather (Eyre et al., 2019) and

climate (Stengel et al., 2020). Often the first step in measuring a geophysical parameter is to determine if the scene is 'clear'. This process is called cloud masking. Cloud masks are needed to observe and retrieve surface parameters. Masks are required



to differentiate clouds from aerosols when running property retrievals or flux calculations. This data is integrated into climate models to generate reanalysis products such as ERA5 (Hersbach et al., 2020), as starting points for future climate modelling (Eyring et al., 2016) or for validating model outputs (Jian et al., 2020), making them an important tool in understanding the
Earth's climate.

However in many clouds masking algorithms high aerosol optical depth events such as dust storms and biomass burning events are also mis-labelled as cloud. This maybe appropriate if the aerosols would cause significant errors in the retrieval. For example regions of high dust loadings are identified and removed so as not to bias sea surface temperature retrievals (Merchant et al., 2006). However the application of satellite data, for example during fire events, may be to identify the thick aerosol
plumes in order to understand the air quality. In this case, removing the thick plumes would adversely affect the intended application.

Clouds and aerosols are observed by both active satellite instruments such as Cloud-Aerosol Lidar and Infrared Pathfinder Satellite Observation (CALIPSO) (Winker et al., 2004) and passive polar orbiting satellite instruments such as the Sea and Land Surface Temperature Radiometer (SLSTR) (Coppo et al., 2013), the Advanced Very High Resolution Radiometer (AVHRR)
(Stengel et al., 2020) and the Moderate Resolution Imaging Spectroradiometer (MODIS) (Justice et al., 1998). Different types of satellite instrument have advantages and disadvantages. Geostationary satellites such as Himawari-8 (Bessho et al., 2016) and the Geostationary Operational Environmental Satellite (GOES) (Schmit et al., 2005) allow for large-scale observation of atmospheric variables with high temporal resolution, medium spatial resolution and good regional coverage. Active instruments, such as the lidar instrument CALIOP on board the polar-orbiting CALIPSO satellite, use on-board radiation sources to
analyse the atmosphere which enables higher horizontal and vertical resolution atmospheric profiles than measurements from passive imagers. However active instruments typically only have a nadir view with a small footprint, significantly reducing their spatial coverage and temporal resolution compared with passive viewing satellites.

In this paper we develop a neural network (NN) cloud mask which, while needing to perform well in all atmospheric scenarios, is specifically focused on minimising the misidentification of thick aerosol as cloud. In the following sections we
first review existing cloud masking techniques, then we describe the approach to collocate the active and passive instruments in order to train a NN algorithm. We describe the structure of the neural net algorithm and demonstrate the algorithm performance by analysing 4 case studies, and comparing qualitatively and quantitatively the performance of the mask with existing cloud masking algorithms from the Australian Bureau of Meteorology (BoM) and the Japanese Meteorological Agency (JMA) (Imai and Yoshida, 2016). The case studies and validation will present results for the cloud mask globally but the case studies will
focus on distinguishing between aerosol and cloud, particularly high aerosol loading events.

## 2   Review of existing cloud identification techniques

Many property retrieval algorithms for passive imagers rely on empirical cloud masks to identify cloud in satellite scenes (Lyapustin et al., 2018). These masks use models of what clouds and aerosols should look like under certain conditions as tests, which are often applied in combination to satellite scenes to identify any cloud and aerosol pixels (Zhu et al., 2015;





Imai and Yoshida, 2016; Le GLeau, 2016). The most significant problem with these algorithms is the dependence on a number of predefined thresholds, which leads to a high sensitivity to surface type, atmospheric variability or instrument calibration and does not capture the more complex, often non linear relationships between instrument channels, clouds, aerosols and the surface. This can result in misidentification of thick aerosol plumes or bright surfaces as cloud, or incorrectly identify thin cloud as aerosol. Bayesian statistical techniques can improve the speed and accuracy of classification, as well as removing the

dependence on arbitrary thresholds (Uddstrom et al., 1999; Hollstein et al., 2015). However, these algorithms often struggle to capture the more complex relationships between the inputs and classifiers as they cannot fully adapt to more complex classifications, such as identifying low, cold cloud over the Poles (Poulsen et al., 2020).

Machine learning techniques, such as NNs, are an alternative to conventional cloud-identification algorithms. These algorithms develop relationships between their inputs and outputs that best fit the data they are trained on, which removes any

dependence on empirical thresholds and allows them to discover complex relationships between multiple input variables, potentially improving the accuracy when compared to empirical and Bayesian algorithms. Examples of neural net algorithms can be found in (Sus et al., 2018; Yao et al., 2018; Mahajan and Fataniya, 2019; Poulsen et al., 2020). The disadvantage of these algorithms at least for supervised neural nets, is they require large amounts of accurately labelled data. This can be achieved by labelling areas as cloud or aerosol by hand as in (Hughes and Kennedy, 2019). However, this is limited by a person's ability

to accurately differentiate clouds from aerosol plumes in satellite scenes and is very time-intensive. An alternative, and the method used in this paper is to collocate the passive imagers with high resolution vertically resolved active instruments, such as radars and lidars, to accurately label pixels which contain cloud or aerosol (Holz et al., 2008; Taylor et al., 2016). This leads to an accurate training and validation set and provides detailed information about the composition of the atmosphere within these pixels. However, the variability of cloud scenes, the narrow swath and long return-to-target times of active instruments

requires a significant number satellite scenes to be processed in order to gather enough data points to effectively train a NN.

## 3 Description of key data sets

Two sources of data were collocated to generate the training and validation data: the AHI instrument on-board Himawari-8 and the CALIOP instrument on-board CALIPSO. The individual data sets are described in the following sections. The collocated data were created with the intention of identifying clouds and aerosols independently in AHI pixels using the active lidar

instrument data as the truth label. While many individual data sets were collocated, not all were included in the final algorithm. The data set was split with 80% used to train the algorithm and 20% of the collocations used to validate the results.

### 3.1 Himawari-8 AHI

The Advanced Himawari Imager (AHI) on-board the JMA's Himawari-8 geostationary satellite is a next-generation passive imager (Bessho et al., 2016). The satellite was launched in October 2014 and became operational in July 2015. AHI has 16

bands covering the visible short-wave (SW) to thermal long-wave (LW) at 0.5-2km resolutions at nadir. The channel specifications are summarised in *Table 1* along with the primary objective of the channel. Full-disk scenes are centred at 140.7° E on the





equator with a field of view (FOV) of approximately 80° in radius and are generated every 10 minutes. The AHI L1b product in Himawari Standard Data (HSD) format is used for this collocation and read in using the Satpy python package (Raspaud et al., 2021) using the default calibration. For the higher resolution channels (see *Table 1*), the values are downsampled to 2km
resolution. The mean and the standard deviation over the downsampled area are calculated and considered as inputs into the NNs. The auxiliary information from AHI is also included in the collocated data, such as the latitudes, longitudes, solar and observation angles.

## 3.2 CALIOP

CALIPSO is an active LIDAR instrument on-board NASA's polar-orbiting Calipso satellite (Winker et al., 2004). CALIOP
uses a dual-band laser to take vertical profiles of the atmosphere directly below its orbital path every 333m along-track. This is reprocessed to 1km and 5km along-track for higher accuracy at the cost of spatial resolution. The instrument has a high sensitivity to the presence of clouds and sensitivity to particle shape and absorption characteristics so it is able to distinguish clouds (liquid and ice) with high accuracy and between aerosol type with limited accuracy. CALIOP provides day and night global coverage albeit with a narrow swath and long return cycle, making CALIOP useful for analysing cloud masks. Each
overpass is made up of profiles containing layers with vertical resolutions dependent on the altitude of the layer that is measured by CALIOP. Both the 5km and the 1km were assessed for the cloud masking algorithm. While the 5km algorithm is more sensitive to optically thin clouds, after initial investigation optimum results were found using the 1km L2 cloud-layer version 4.20 product (CAL_LID_L2_01kmCLay-Standard-V4-20) because of the increased sensitivity to small clouds.

The variables from CALIOP contained in the collocated data include the vertical feature mask, feature top altitudes and
105 cloud-aerosol discrimination (CAD) score are saved in the collocated data file for each layer to allow for analysis of clouds and their properties.

## 4 Methodology

In order to create the neural net algorithms we need to collocate AHI data with the most accurate 'truth' data from the CALIOP instrument. To ensure the most applicable training data is generated, we need to take into account parallax when carrying out
the collocations between the active and passive satellites . The collocated data set was then used to train and validate the neural net algorithms which are described in *Sect. 4.2*.

### 4.1 Collocation of Data

The overall collocation process is shown in *Fig. 1*. The collocation process has two main steps. First a rough collocation to identify temporally matching Himawari and CALIOP orbits and a second step to perform a parallax correction.
The initial rough collocation, shown in panel a of *Fig. 1* follows the steps below:

- The section of a full CALIOP overpass that lies within the FOV of AHI is identified; leads to temporal uncertainty of approximately ±5 minutes.





– The AHI scenes that lie with the start and end of the CALIOP section run are identified.

– The AHI scenes are retrieved.

### 4.1.1 Parallax correction

Once the initial collocation is performed, the CALIOP subsection that lies within the start and end of each AHI scene scan time is found. From these sections, every CALIOP layer was parallax corrected to ensure accurate collocation with its corresponding AHI pixel. This is an important step for making sure the dataset was accurately label, as shown in *Fig. 2*, the parallax between the 2 instruments can cause layers to be shifted by several pixels in AHI from their position as observed by CALIOP, this can be particularly significant for high clouds at the edge of the full disk. The process for performing the parallax correction is shown in panel c of *Fig. 1*.

The parallax correction for each layer was performed by:

– Calculating the observation angles of the layer as seen by AHI at the position and altitude specified in the CALIOP data.

– The observation angles of the CALIOP layer as seen by AHI was then matched with the observation angles for AHI corresponding to the Earth's surface.

– Where the observations matched, the layer was assigned to the collocated AHI pixel; as the match is to the closest pixel, this leads to a spatial uncertainty of approximately $\pm1$km as nadir for AHI.

– This was repeated for every layer and a pseudo-CALIOP profile was generated for each pixel. This includes thin layers that AHI may struggle to observe and are accepted as a potential source of error in the final cloud mask.

After each CALIOP file had been collocated, the data was cleaned by:

– Checking for duplicate layers and keeping only the closest matching AHI pixel.

– Removing layers with AHI pixel data or CALIOP layer information missing.

These checks ensure the quality of the data in the final product is high. No further quality flags were applied to the collocated data, but were instead stored with each entry to allow for further filtering of the collocated data to ensure only high quality data is used to train the algorithm.

Approximately 7300 CALIOP overpasses were collocated for 2019 to make the full training and test validation data set, making this data suitable for large-scale statistical studies of cloud and aerosol in AHI scenes. The density distribution of collocated AHI pixels for all of 2019 is shown in *Fig. 3*, showing a relatively even distribution across the AHI's field of view (FOV) for day and night. As can be seen in panels a and c of *Fig. 3*, outside the polar regions, there is very little regional bias in the collocated data for day and night pixels, with the notable exception of the region where AHI's twice daily downtime syncs with CALIOP's orbit. However, due to Calipso's sun-synchronous orbit, twilight data is only available towards the Poles. The concentration of collocated data at the poles for twilight scenes should be kept in mind when analysing the results.



## 4.2 Training and Validation of Neural Networks

The cloud identification algorithm is based around a feed forward neural net and implemented using the Tensorflow (version 2)
package for Python (Abadi et al., 2016). Three models were created for day, twilight and night scenarios. The time of day was
separated by solar zenith angle (SZA) and corresponding to $0° \leq SZA < 80°$, $80° \leq SZA < 90°$ and $SZA \geq 90°$ respectively.
After experimenting with different input data the final inputs to the neural net are:

- **Day -** Channels 1 - 16 (mean values for down-scaled channels), Himawari observation angles and cos(SZA).

- **Twilight -** Channels 1 - 16, including standard deviations and mean values for down-scaled channels.

- **Night -** Channels 7 - 16 and Himawari observation angles.

For each of the different networks the collocated data is normalised between approximately -1 and 1 to ensure optimal neural
net performance using the following simple equation:

$$I_{norm} = \frac{I - I_{avg}}{I_{max} - I_{avg}} , \tag{1}$$

where $I$ is the original input value in percentage reflectance, brightness temperature and degrees for shortwave bands,
longwave bands and angles respectively, $I_{norm}$ is the normalised input value, $I_{avg}$ is selected by the approximate middle of
the range of physically expected values and $I_{max}$ is the maximum expected input value. $I_{avg}$ has values of 50, 273.15, 0 and 0
for shortwave bands in percentage reflectance, longwave bands in brightness temperature, observations angles in degrees and
cos(SZA) respectively. $I_{max}$ has values of 100, 423.15, 90 and 1 for shortwave bands, longwave bands, observation angles and
cos(SZA) respectively. The CALIOP data is labelled using binary flags. The binary flags for cloud identification was selected
from the collocated CALIOP profiles. If the top layer of the collocated CALIOP profile was found to be cloud, a filter would
be applied to decide the label. If the layer has a high CAD score (CAD score > 50 (Koffi et al., 2016)) or is above 7.9km in
altitude, the layer is called cloud. The CAD score filtering ensures any thick aerosol plumes from biomass burning that are
occasionally flagged as cloud by CALIOP are not erroneously labelled as such for the NN training. The altitude threshold is
taken from Labonne et al. (2007) with the value chosen to be slightly higher than the maximum injection height of biomass
plumes from this study. This minimises the number of true cloud layers that are mis-labelled as non-cloud in the training of the
NNs.

Each NN is optimised by iterating through 1, 2, 5, 10 and 20 layer structures. For each iteration, 10, 20, 50, 100 and 200
neurons were used per layer, e.g. 2 layers of 50 neurons or 5 layers of 100 neurons. All available inputs were used and a
subset of 2000 collocated datasets with a QA filter of 0.7 on collocated overpasses were used for the optimisation process.
This was repeated 5 times for the stochastic gradient descent (SGD) and Adam (Kingma and Ba, 2015) optimisers to account
for different cloud and cloud-free scenarios and also to account for the statistical effect due to the small variations in the
minimisation process. In total, this corresponds to 50 configurations that were investigated. The simplest NN structure with the
highest average area under the receiver operating characteristic (ROC) curve value was used as the final algorithm and applied





to the full dataset. This is found to be 10 layers of 100 neurons each for the day, twilight and night algorithms when using the
SGD optimiser. Therefore the final structure of each NN is:

- An input layer of N neurons, where N is the number of inputs used, with tanh activation.

- A dropout layer with dropout rate of 0.2 to prevent overfitting (Srivastava et al., 2014).

- 10 layers of 100 neurons each with tanh activation for each layer.

- Single neuron output layer with sigmoid activation.

The NNs give a continuous output between 0 and 1, so an optimal threshold must be found for each NN to convert the values to
a binary cloud mask. These are found after the optimisation process by taking the threshold which maximises the true positive
rate (TPR) and minimises the false positive rate (FPR) of the final algorithms. For this analysis, both continuous and binary
values are used.

For validation of the NNs, several metrics are used. The false positive rate (FPR) and true positive rate (TPR) can be
calculated by applying thresholds on the continuous output of the NNs. The binary outputs can also be processed into values
such as the Kuiper skill score (KSS), which is calculated by,

$$KSS = TPR - FPR = \frac{a}{(a+c)} - \frac{b}{(b+d)}, \tag{2}$$

where $a$ is the number of true positives, $b$ is the number of false positives, $c$ is the number of false negatives and $d$ is the
number of true negatives (Hanssen and Kuipers, 1965). KSS is useful metric that balances the TPR and FPR for analysing
how accurate an algorithm is at identifying cloudy pixels from non-cloudy pixels and comparing different products. A perfect
KSS is 1, representing a perfect classifier algorithm, whilst 0 indicates an algorithm with no skill. Using statistical analysis
alongside case studies allows for a wide-ranging analysis of the NN algorithms.

A subset of 30 collocated overpasses from 2020, which is not used in the initial training and validation, is used to carry out
a statistical analysis of the NN mask, including a comparison with cloud masks from the JMA and BoM described briefly in
Sect. 4.3. The masks are also compared for different case studies.

### 4.3    Description of auxiliary cloud masks

The finalised NN mask is compared to cloud masks from the BoM and JMA. The cloud masks used in the comparison are
described in *Sect. 4.3.1* and *Sect. 4.3.2*.

### 4.3.1    Bureau of Meteorology (BoM)

The BoM mask uses the NWCSAF cloud masking algorithm run with NOAA Global Forecast System (GFS) (NOAA, 2021)
NWP information. The full description of the algorithm is described in Le GLeau (2016) and the key features are outlined
below.





- The algorithm is based on look-up tables (LUTs) to calculate thresholds which are applied to individual pixels to identify cloud.

- The algorithm is split into land and sea algorithms for an initial run. These algorithms use a combination of thermal channel differences during day, twilight and night, as well as the $0.6\mu$m channel during day and twilight.

- A range of other threshold tests are applied to further identify features which are usually difficult to identify, e.g. low-level cloud, cloud edges, thin cirrus.

### 4.3.2 Japanese Meteorological Agency (JMA)

The JMA mask is based on both the NWCSAF Cloud Mask (CMA) (Le GLeau, 2016) and NOAA Advanced Baseline Imager Clear Sky Mask (ACM) products (Heidinger and III, 2013). The full algorithm description can be found in Imai and Yoshida (2016).

- The main concept behind the JMA cloud mask is to compare AHI channel data to Numerical Weather Prediction (NWP) values to decide if a pixel is cloudy.

- A subset of 9 channels are used (Bands 3,4,5,7,10,11,13,14,15)

- Snow/sea ice are filtered and flagged in final cloud mask.

- Aerosol pixels are also filtered using the ash detection algorithm from Pavolonis et al. (2013).

- The cloud mask utilises information from the previous scene and a scene one hour previously to capture temporal variation.

## 5 Results

### 5.1 Statistical comparison of cloud masks using CALIOP

The cloud masks were compared statistically by using collocated CALIOP data as the truth label. 30 collocated datasets that were not used for training and validation were analysed by the NNs. This dataset consisted of approximately 60,000 collocated pixels randomly selected from January 2020 to June 2020. The pixels were also collocated with the matching BoM and JMA
binary cloud masks. An example of collocated data used in this comparison is shown in *Fig. 4* with the cloud masks and truth label shown above the profile curtain plot to demonstrate the strengths and weaknesses of each cloud mask. It can be seen that the NN and JMA cloud masks show similar performance whilst the BoM mask classifies most pixels either side of clouds as cloud, leading to a conservative cloud mask. These performances are further demonstrated in the statistical analysis.

The TPR and FPR for each cloud mask was calculated and plotted on a ROC curve representing the combined outputs of
235 the NNs in *Fig. 5*. An overall comparison of the performance shows that the NN cloud mask has the best performance over all,





with a KSS of 0.691 versus 0.589 for the JMA product and 0.472 for the BoM product. The low FPR and low KSS of the JMA product suggests a cloud mask that balances cloud masking without being conservative, i.e. classifying non-cloud pixels as cloud to ensure all cloud is cleared in a scene, but does not perform as well as the NN, whilst the high FPR and low KSS of the BoM mask suggests a cloud mask that over-classifies pixels as cloud. The combination of high TPR and high KSS shows that the NN is better at not classifying non-cloudy pixels as cloudy than the JMA and BoM masks. This is important for accurately identifying optically thick aerosol plumes.

The continuous output of the NN mask can have any threshold applied to produce a binary mask, e.g. all pixels with continuous values greater than 0.5 can be assigned as cloud. The values presented were found after optimisation, but the threshold can be adjusted so that it matches the TPR or the FPR of the other algorithms. In this analysis, the threshold that yields the equivalent TPR of the BoM and JMA algorithms separately was calculated and are represented by the dashed coloured lines in *Fig. 5*. The blue and magenta horizontal lines correspond to the TPRs of the BoM and JMA masks respectively, whilst the same coloured vertical lines show the FPRs of the of the NN mask at those TPRs. If the NN's thresholds were chosen so that the TPR of each algorithm matched, the associated FPRs would be 0.314 versus 0.464 for NN and BoM algorithms and 0.106 versus 0.198 for the NN and JMA algorithms respectively. This implies that the NN accurately identifies 1.11 and 1.28 times as many non-cloudy pixels when compared to the BoM and JMA masks respectively if the binary output threshold is tailored to match their TPRs.

The performance for each mask was further broken down by satellite zenith angle, solar zenith angle, airmass and surface type. The results show that the NN algorithms perform best during day and night but lower performance during twilight, where the FPR increases significantly over the day and night algorithms. Overall, the NN mask shows a notable improvement over the BoM and JMA masks, which show lower values of KSS for all situations. From the TPR and FPR values presented in *Table 2*, the JMA mask generally has similar TPR values when compared to the NN in all scenarios, but with higher FPR values, whereas the BoM mask has significantly higher TPR and FPR values. This shows that the JMA and BoM masks are more conservative than the NNs, i.e. the NNs are better at accurately identifying non-cloud during all conditions. The NN mask also outperforms the other masks at all satellite zenith angles, retaining similar TPR and FPR values over all angles, and does not drop in performance towards the edge of the scene at the same rate as the BoM and JMA masks. In particular, the BoM FPR values increase significantly with satellite zenith angle, leading to very poor performance at high angles. All the algorithms show approximately equivalent changes in performance for airmass values above 3, but the NN mask does not have as large a drop-off at low airmass values.

## 5.2 Explainable machine learning

Machine learning models are often seen to be opaque black boxes, from which it is difficult to extract physical interpretations. However there exists new tools that can begin to explain the performance of individual machine learning models. SHAP (Lundberg et al., 2020; Lundberg and Lee, 2017) is one such tool. The package is based on the Shapely game theory approach to explain multi-input models. SHAP works by perturbing each model input over several iterations and calculating how much this affects the final output of the model. From this, a value related to the importance of the input can be extracted for the





model. Carrying out this analysis for a range of pixels, both cloud and non-cloud, allows the otherwise opaque models to be
analysed statistically. In addition using this tool it is possible to identify the importance of each input for an individual pixel.

The final versions of the day, twilight and night NNs that make up the NN cloud mask are analysed using the SHAP package.
As the SHAP computation is computationally expensive, the results of averaging 5 different subsets of 200 randomly sampled
pixels per iteration for each NN over land and ocean are presented in *Fig. 6*. The error bars are generated by taking the standard
deviation from the sample of 5 subsets.

These results show that during the day, for both land and ocean, the NN relies on information found in the visible and
near-visible short-wave infrared (SWIR) mean values, as well as channels in the long-wave infrared (LWIR), with band 15
(12.4μm) being particularly important due to a small amount of water vapour absorption compared with the 'cleaner' band 13
(10.4μm) channel. The high importance of the SW channels is due to the spectral response of clouds in the visible and SWIR
bands, which differs compared to most surface types, especially over the dark ocean surface. From *Fig. 6*, the relatively high
significance values for the visible channels show that they are particularly important over land, where it can be seen that the
blue (0.47μm) band - band 1 - is approximately as important as band 2 (0.51μm) and almost as important as band 3 (0.64μm).
Furthermore, the different spectral responses of cloud and aerosol in the blue band are likely why the NN shows a notable
improvement over the other algorithms, especially during high aerosol loading events as those that will be shown in *Sect. 5.3*.
Bands 1 and 2 are not used in the JMA and BoM clouds masks so maybe one of the key reasons for the improved performance of
the NN algorithm. The dependence on LWIR atmospheric windows is due to the difference in brightness temperature between
the surface and cloud, particularly at higher altitudes. In addition, the relatively high importance of band 11 (8.6μm), which
contains information on cloud phase (see *Table 1*), suggests the NN is attempting to extract information on the microphysical
properties of a pixel from this band and may be a key factor in the improvement of the NN dealing with optically thick aerosol
plumes when used in combination with the visible bands. The relatively high dependence on the observation angle implies that
the NN is accounting for aberration and path-length effects on the spectral response of clouds at the edge of a scene. The solar
zenith angle during day and other water vapour channels were not found to be important.

For night pixels, it can be seen that band 15, the thermal window channel at 12.4μm, dominates the classification skill over
ocean, whilst the importance of all the channels is much more evenly distributed for land pixels. Band 15 is important for
detecting low-level moisture and can be useful in identifying low-level cloud that might be of a similar temperature to the
sea surface temperatures in other channels, leading to the high importance over ocean. Over land, the larger range in surface
temperature values at night leads the NN to depend on several inputs rather than a single atmospheric window channel. The
relatively high importance of band 7 (3.9μm), which has a SWIR and LWIR component, suggests the NN is also extracting
information on low-level cloud from this band. Observation angles are also important for the night algorithm likely due to the
path length of radiation in the atmosphere, as well as potentially being used as a proxy for identifying high latitude regions.

Twilight pixels are a challenge for cloud masks due to the very high airmass values causing visible channels to appear
significantly different from the day equivalents. The twilight NN appears to place lower importance on the shorter wavelength
visible channels, which are more strongly scattered during twilight, and instead relies more strongly on the band 4 (0.86 *μm*)
mean, the SWIR and the thermal channels. Over ocean, the NN places very high significance on the band 4 mean, which is less





affected by Rayleigh scattering and is dark for the ocean surface. This suggests the algorithm looks for high band 4 values to
identify cloud over ocean in combination with the traditional infrared window channel of band 14. Over land, bands 4, 10 and
14 have approximately equivalent significance in the NN. Bands 4 and 14 serve the same role over land as they do over ocean.
However, unlike over ocean, some land surface types can be bright in band 4 at twilight. This causes the NN to require a water
vapour absorption band to effectively identify cloud over land during twilight and the NN has found band 10 to be most useful
for this purpose.

### 5.3 Case Studies

In this section we present a full disk comparison, a case study over China during a significant dust storm and 3 regional case
studies over Australia.

### 5.3.1 Full disk: Global view

The case study presented in *Fig. 7* is a full disk scene for the 1st of November 2020 03:30 UTC. The scene is predominantly
day light, with twilight just appearing in the top right of the image. A region of sunglint can be seen north of Australia.

All the case studies have the same format, with panels a, c and e showing true colour, natural colour and EUMETSAT-style
dust RGB false colour composites. The true colour RGB false colour composite is generated from the visible bands (bands 1-3)
to show approximately how the scene appears if observed with the human eye. The natural colour RGB false colour composite
uses bands 2-4 and shows vegetated areas as green, arid areas as pale brown, cold ice-phase cloud as pale blue and warmer
water-phase cloud as white. Both the true and natural colour RGB false colour composites show sunglint due to the presence
of short-wave bands. The EUMETSAT-style dust RGB false colour composite is derived from bands 11, 13 and 15 and does
not show any sunglint due to being purely thermal channels. This RGB false colour composite can be used at night and is
particularly useful as it shows cold cloud as black and red, including thin cirrus, and shows thick dust plumes as purple. In
panels b, d, and f, results from the cloud masks are shown. In panel b, the continuous output from the NN mask is shown for
the scene. In panel d, the binary cloud mask from the NN is shown. Finally, in panel f, a comparison between the NN, BoM and
JMA masks is shown. In this comparison, dark red and dark blue corresponds to where all masks agree pixels there is cloud
and non-cloud respectively. The pale red sections are where the BoM and NN masks agree, but the JMA mask gives a different
classification, e.g. BoM and NN say cloud but JMA says non-cloud. Similarly, white is where JMA and the NN agree and pale
blue is where JMA and the BoM masks agree. From the NN binary mask and the mask comparison, the class assigned by the
JMA and BoM masks can be inferred.

Three different RGB false colour composite false colour images are presented which highlight different features in the scene,
such as ice-phase clouds which appear cyan in the natural colour RGB false colour composite. Both the continuous output from
the NN algorithm and the binary mask are shown in panels b and d of *Fig. 7* and show that regions of small clouds and cloud
edges are the most uncertain for the NN mask. Panel f of *Fig. 7* shows the comparison between the NN, BoM and JMA cloud
masks. All three masks show good agreement for the majority of pixels, with areas of high cold cloud shown as red and clear
land and ocean broadly shown as blue. Generally, areas where cloud masking appears to fail are towards the edges of clouds





and clouds at small-scales. Towards the north of Australia is a region of sunglint. Some small areas within this region are shown as pale blue in the mask comparison, indicating that the JMA and BoM cloud masks agree on the classification of this region

but the NN mask disagrees with the other masks. In the NN mask it can be seen that the area of sunglint is broadly shown to be non-cloud i.e the mask is not misclassifying some sunglint pixels whilst the BoM and JMA masks are. In the twilight region of the scene, which is outside the collocated region of twilight data, it can be seen that the NN cloud mask fails to detect low level cloud. This is likely due to low level cloud outside the collocated region appearing significantly different to within the region. The NN mask likely interprets the cool bright surface as polar surface and such behaviour is expected when using an

algorithm significantly beyond its training data set. Therefore the twilight algorithm does not perform well outside the region it was trained on and aiming to improve this algorithm will be an area of future development.

### 5.3.2   Dust Storm over China

The scene presented in *Fig. 8* shows a significant dust storm over China from 15/03/2021 03:30 UTC. In panel e of *Fig. 8*, the dust plume can be seen as the pink region across China towards the north of the centre of the scene and is partially covered

by cloud. In addition, there is a region of snow and ice towards the north of the scene which is also covered by some cloud. From panels d and f of *Fig. 8*, it can be seen that the NN cloud mask improves on the JMA and BoM masks when dealing with snow and ice in the north of the scene as it does not misclassify the majority of the snow and ice. However, all the masks fail to effectively classify the dust plume, with the exception of the NN mask accurately classifying a small section of the dust storm to the north of the Korean peninsula. Given that this event was a historically significant event with an unusually high

plume (Filonchyk, 2022), the failure of the cloud masks might be expected. In particular, it shows that the NN cloud mask is only as effective as its training data and extreme events that it is not trained for will cause the mask to fail, although under more extreme scenarios than the JMA and BoM masks. In panel b of *Fig. 8*, pleasingly it can be seen that the section of the dust plume that is towards the centre of the scene is assigned scores significantly below values given to clouds - the plume has values of approximately 0.5, whereas clouds have values close to 1 - indicating that the NN mask is not confident the plume is

cloud. A future algorithm could use this information within a convolutional NNs to improve the performance for large plumes or to develop uncertainty metrics.

The following case studies look at a mixture of high AOD events, small/subpixel cloud and sunglint in scenes from Australia. The regions each of the case studies cover is show in panels a and b of *Fig. 9*.

### 5.3.3   Australian case study A: smoke from the 2019-2020 Australian bushfires

Case study A is a region on the south east coast of Australia from an AHI scene on the 1st January 2020 at 03:30 UTC during the Black Summer bushfires (see *Fig. 10*). This scene shows an optically thick smoke plume as well as thick and optically thin cloud. This scene shows the difficulty in distinguishing between smoke and cloud. The majority of the smoke plume is successfully classified as non-cloud by the network. When compared to the other cloud masks, it can be seen that the network classifies less of the smoke as cloud than the JMA mask and both classify significantly less smoke as cloud than the BoM

mask, which classifies optically thin smoke as cloud. There are regions off the coast towards the centre of the scene that are





classified as cloud by the NN and appear to be low, warm cloud under optically thick smoke. This is not surprising as the spectral signature is predominantly sensitive to the top layers of aerosol or cloud.

The overall improvement in performance for the NN mask is likely due to including the additional channels not included in the JMA or BoM masks. In particular for distinguishing optically thick smoke during the day we expect the 0.47µm channel to add useful information, as the spectral response of cloud and smoke are significantly different in this band (Gautam et al., 2016). However, there are small regions of the smoke plume over land and along the coast which are classified erroneously as cloud by the NN. From the continuous mask, it can be seen that these regions are classified as cloud by the NN mask with relatively high confidence. Parts of these regions correspond to cloud which can be seen in the natural colour RGB false colour composite, but not all of it. Comparing the areas that do not appear to have cloud with the dust RGB false colour composite show that it aligns approximately with regions of dense smoke that have a pink colour suggesting that the smoke plume has significant dust contamination. In addition, these regions may contain particularly high levels of water vapour, which from *Sect. 5.2* is known to be important in cloud classification for the NN. This water vapour likely comes from the intense period of fire activity from just before this case study (Peterson et al., 2021). There is the possibility that the water vapour has begun to condense within these regions of the plume and therefore could be forming cloud particles. However, without the ability to eliminate the presence of cloud beneath this layer, it is not possible to determine the exact cause of this issue and further investigation is warranted.

### 5.3.4 Australian case study B: Northern Australian mixed scene

This case study is also taken from 1st January 2020 at 3:30 UTC and covers the Gulf of Carpentaria and Northern Queensland (see *Fig. 11*). The region illustrates many scenarios that are typically difficult to identify with cloud masks. Present in this scene is sunglint in the top left, sediment in the water off the coast, thin cirrus at the top of the scene and sub pixel cloud over the land.

In this region, high, cold ice-phase cloud is consistently flagged as cloud by all the masks. However, both the JMA and BoM masks falsely classify more non-cloud pixels as cloud than the NN mask does over ocean. Towards the left of the scene, the JMA and BoM masks misidentify areas of sunglint as cloud. Both the JMA and BoM masks also identify the edges of areas of murky water off the west coast of Cape York as cloud. Over land, the low, warm, small-scale clouds over Cape York are not well identified by the NN, whereas the JMA and BoM masks flag more of this areas as cloud.

### 5.3.5 Case study C: Central Australian optically thick dust

*Figure 12* shows a region in central Australia with a thick dust plume from 5th January 2020 at 5:00 UTC. This plume is accurately classified by the NN, and the majority of the the plume is correctly classified by the BoM mask. However, both the BoM and JMA misclassify some or all of the plume as cloud. To the north of the dust plume, there is a salt lake visible in the image. This is classified by the NN as non-cloud, but misclassified by the other masks. This indicates another area of improvement for the NN and is particularly important for retrievals over Australia where salt lakes are prevalent.



Towards the center and top of the scene there are areas of small-scale cloud. The JMA and BoM masks identify the majority of the cloud in this region, whereas the binary NN mask fails to identify the smallest clouds. However, it can be seen that this region has values of approximately 0.5 or higher in the continuous mask, indicating that a lower threshold could lead to better classification. The dust plume has continuous values significantly below 0.5, so decreasing the threshold would not lead to misclassification of the dust plume, allowing the NN to be significantly more accurate than the other masks in this scenario.

## 6 Conclusions

Identifying cloud affected scenes in satellite data is an important step prior to retrieving geophysical properties from remote sensing data. Failure to identify cloud correctly can result in substantial biases in a geophysical retrieval. In practice this means that cloud masks are tailored towards conservatively clearing any non-surface features at the cost of identifying many non cloud scenes as cloud.

However conservative cloud basks can introduce other biases. In the cases of air quality and fire applications it is important to remove cloud while still identifying the thick aerosol plumes. In this study, a set of neural networks (NNs) are developed to improve the detection of cloud and aerosol for AHI (Bessho et al., 2016) scenes with the specific goal of identifying cloud affected scenes as accurately as possible and at the same time improving cloud-aerosol discrimination during heavy aerosol loading events.

Data for training and validating these NNs are generated by collocating the AHI instrument with parallax-corrected CALIOP (Winker et al., 2004) 1km cloud-layer products. Approximately 7300 CALIOP overpasses are collocated from 2019 for training separate day, twilight and night algorithms. These algorithms are optimised by sweeping over a range of possible structures using either the stochastic gradient descent or Adam optimisers. The optimal algorithms are then statistically compared to the Japanese Meteorological Agency (JMA) and Bureau of Meteorology (BoM) operational cloud masks (Le GLeau, 2016; Imai and Yoshida, 2016; Heidinger and III, 2013) using 30 collocated CALIOP overpasses from 2020. Analysis of KSS values showed that the NN cloud mask is an overall improvement over the BoM and JMA cloud masks in all scenarios. When using the NN with a true positive rate equivalent to the other masks, the NN accurately detects 1.11 and 1.28 times as many non-cloud pixels versus the JMA and BoM masks respectively.

The NN algorithms were analysed using the SHAP package (Lundberg et al., 2020; Lundberg and Lee, 2017) to understand the influence of each input on the NN performance. For the day algorithm, it is found that all the visible channels are important, including bands 1 (0.47μm) and 2 (0.51μm), as well as the atmospheric window thermal channels. The night NN relies on the atmospheric window channels, as well as band 7 ($3.9\mu m$), whilst the twilight NN relies heavily on the band 4 ($0.86\mu m$), likely due to less Rayleigh scattering in these bands compared with shorter wavelengths, as well as the SWIR channels values to classify cloud, and the traditional atmospheric window thermal channel. However, a lack of collocated twilight data outside the polar regions limits the finding for twilight scenes.

The masks were compared for several AHI scenes covering a range of difficult scenarios, the results suggested the NN mask improves on the JMA and BoM masks for optically thick aerosol plumes, regions of sunglint and challenging surface types.





One case study was a scene that contained a very optically thick smoke plume, taken from 1/1/2020 03:30 UTC during the Black Summer bushfires of Australia. This showed that the NN cloud mask significantly improves on the BoM and JMA mask at classifying smoke as non-cloud. The NN mask struggles to deal with small scale cloud, but due to it's improved abilities to differentiate thick aerosol plumes from cloud, increasing the threshold on continuous values can rectify this issue. In addition, the twilight algorithm struggles outside the polar regions.

Overall, the NN mask shows improved performance over the JMA and BoM cloud masks for all cloud scenes, particularly during heavy aerosol loading events. The findings suggest that including additional channels such as bands 1 and 2 in the JMA and BoM algorithms could aid in the discrimination between clouds and heavy aerosol loadings during daylight conditions.

*Code and data availability.* The code and models needed to run the NN cloud masks is available at https://github.com/dr1315/AHINN. The data used for training and validating the models is available from Robbins et al. (2021).

*Author contributions.* DR led the study, wrote code and analysed results. CP and SRP contributed to code development, SS advised on interpretation of results. All authors contributed to the science aspects. DR prepared the manuscript with contributions from all authors.

*Competing interests.* The authors declare that they have no conflict of interest.

*Acknowledgements.* The CALIOP L2 1km Cloud Layer data were obtained from the NASA Langley Research Center Atmospheric Science Data Center. This research/project was undertaken with the assistance of resources and services from the National Computational Infrastructure (NCI), which is supported by the Australian Government. The AHI data were obtained from archives on NCI's Gadi. The JMA cloud mask was accessed from https://registry.opendata.aws/noaa-himawari.



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



**Table 1.** Description of Himawari-8 AHI channels (Bessho et al., 2016) and purpose (Zhang et al., 2018).

| Band | Wavelength (μm) | Spatial resolution (km) | Comments |
|------|-----------------|-------------------------|----------|
| 1 | 0.47 | 1 | aerosols absorption |
| 2 | 0.51 | 1 | composite imaging |
| 3 | 0.64 | 0.5 | vegetation, aerosol over water |
| 4 | 0.86 | 1 | Cirrus clouds |
| 5 | 1.6 | 2 | phase and particle size, snow |
| 6 | 2.3 | 2 | land, phase particle size and snow |
| 7 | 3.9 | 2 | clouds and night time fog |
| 8 | 6.2 | 2 | high altitude water vapour |
| 9 | 6.9 | 2 | mid altitude water vapour |
| 10 | 7.3 | 2 | low altitude water vapour |
| 11 | 8.6 | 2 | total atmospheric water cloud phase dust |
| 12 | 9.6 | 2 | ozone |
| 13 | 10.4 | 2 | surface clouds atmospheric window |
| 14 | 11.2 | 2 | clouds, atmospheric window |
| 15 | 12.4 | 2 | total water ash, atmospheric window |
| 16 | 13.3 | 2 | air temperature cloud height |





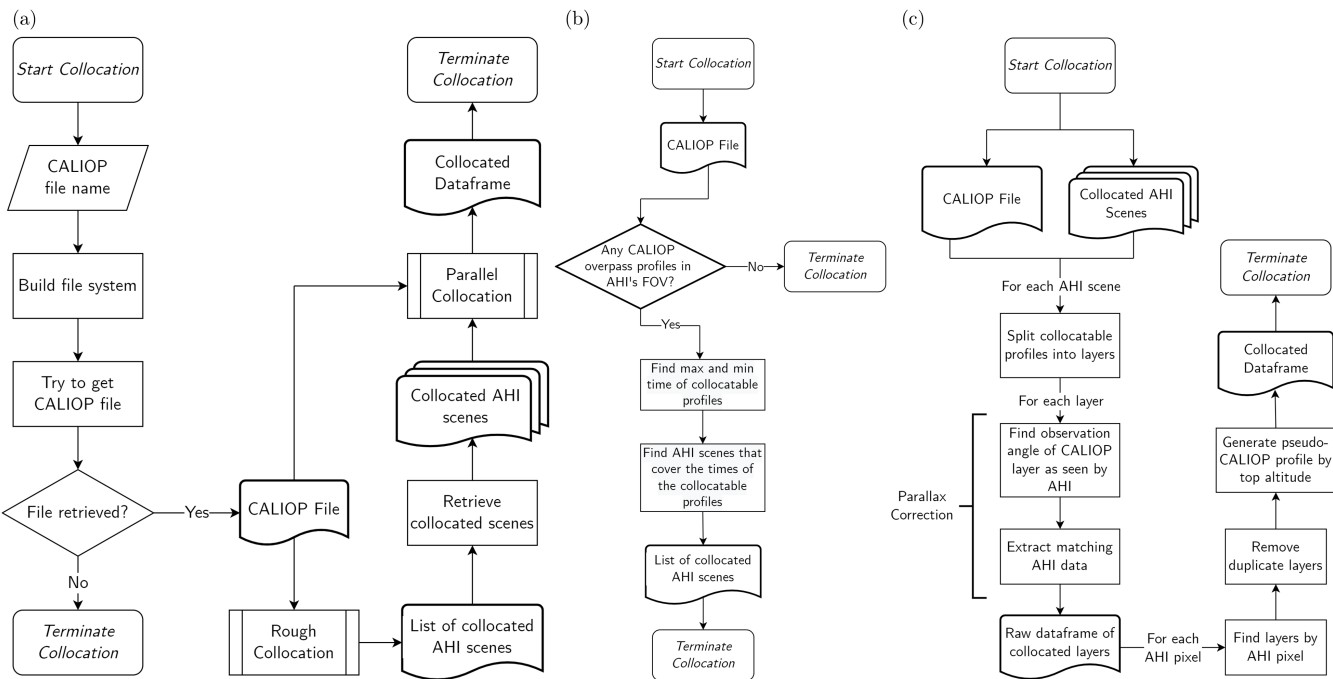

**Figure 1.** The full collocation process for a single CALIOP file. The full process from CALIOP file to collocated dataframe is described in (a), whilst the processes of finding collocated AHI scenes and collocating CALIOP profile layers are described in (b) and (c) respectively.





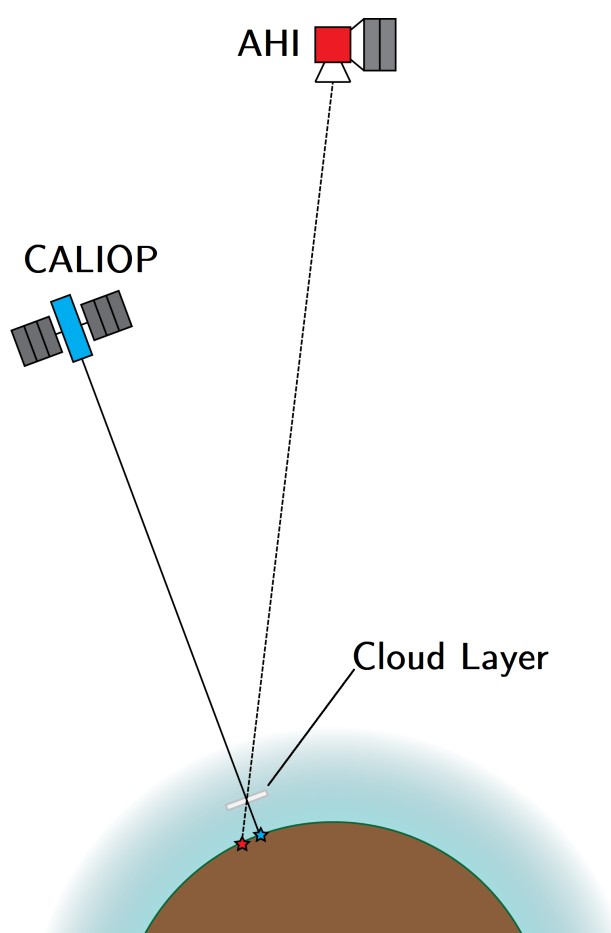

**Figure 2.** The difference in observation angles between CALIOP (blue) and AHI (red) for a layer (grey) at some altitude above the surface of the Earth (brown). At more extreme observation angles for AHI, i.e. towards the edge of the FOV, large differences in the apparent location of the layer can occur. For example, if the elevation angle of AHI from the layer is 45° at an altitude of 10km, the location of the layer is shifted by 10km, approximately the equivalent of 5 pixels at 2km resolution.





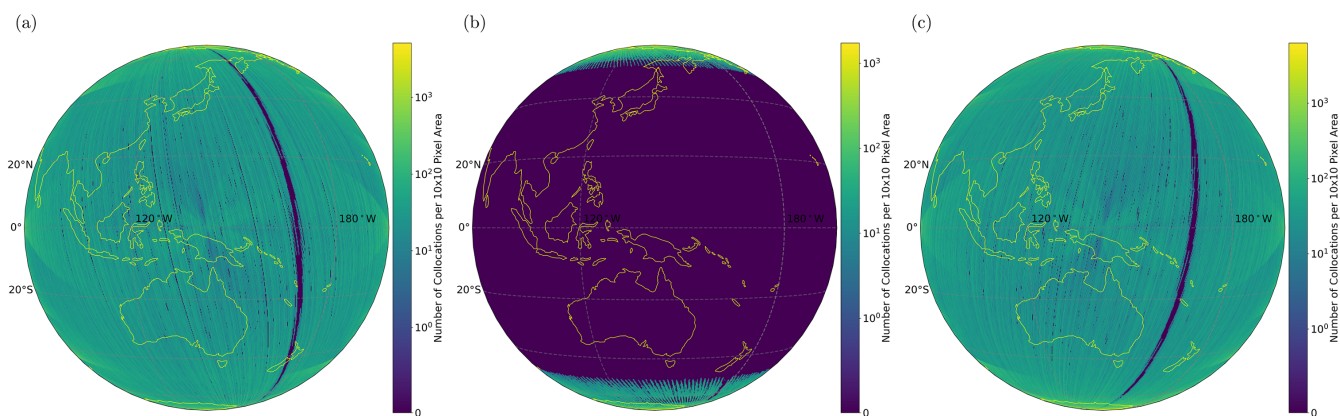

**Figure 3.** Heatmaps of collocated AHI pixels within a 10 pixel by 10 pixel area in the AHI FOV for the full 2019 dataset broken down by (a) day, (b) twilight and (c) night. The northern and southernmost region contain the highest concentration of collocated pixels as CALIOP is in a polar orbit, causing it to return to the same area in each AHI scene. Outside these regions, AHI pixels are relatively evenly distributed across the whole AHI FOV, with the exception of the region to the centre right of the FOV, which is due to AHI having gaps in coverage at 02:40 UTC and 14:40 UTC every day.

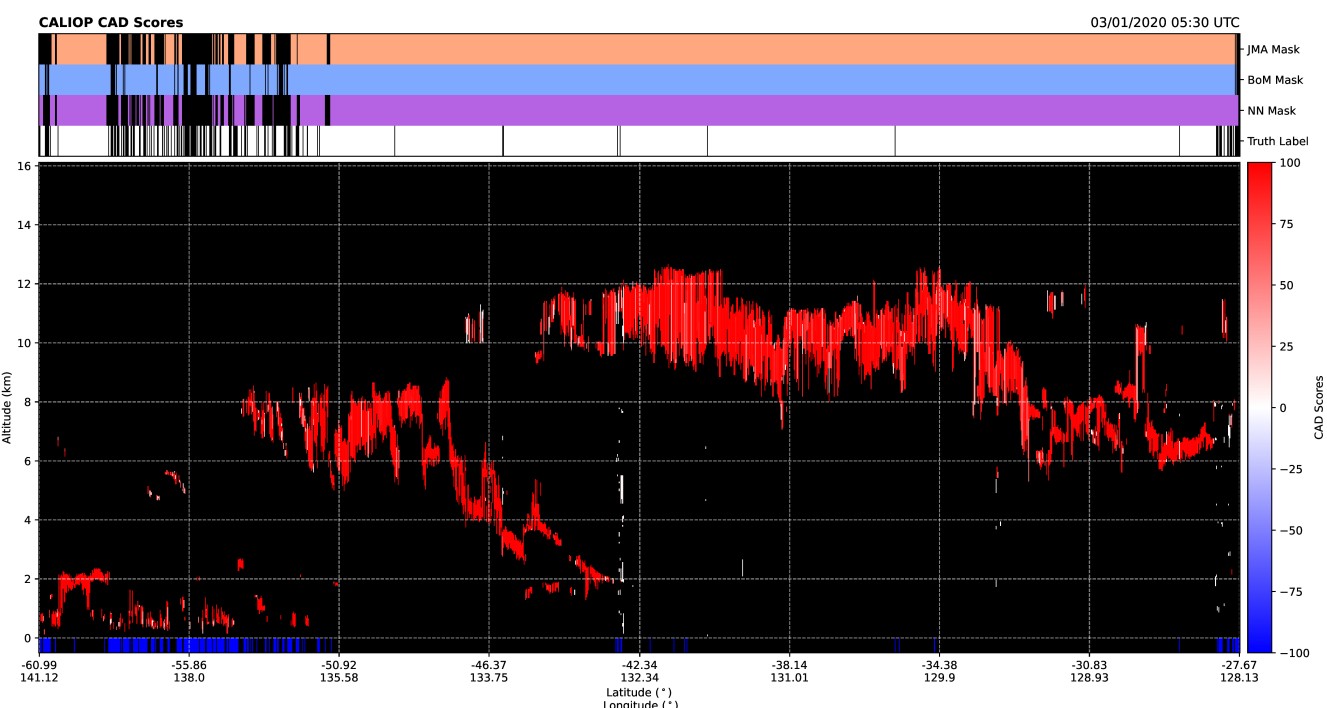

**Figure 4.** The CAD scores from a collocated CALIOP overpass are plotted along with the cloud masks used in this study and the truth label used for training the NN algorithms.



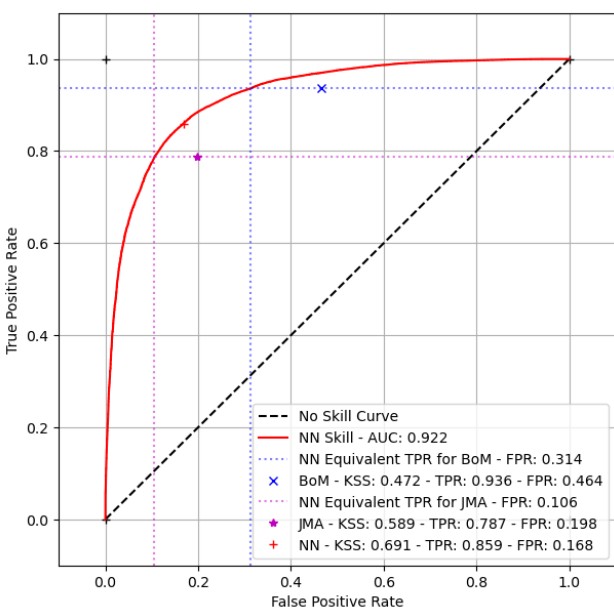

**Figure 5.** ROC curve comparing the false positive and true positive rate of the cloud masks. The ROC curve is produced using the continuous output from the neural network, whilst each point is calculated using the binary cloud mask for each algorithm. The dotted lines indicate where the algorithm point would be if it used a binary threshold that matched the associated algorithms' (denoted by colour) TPR.



**Table 2.** Summary of Kuiper Skill Score (KSS), True Positive Rate (TPR) and False Positive Rate (FPR) for various angular geometries, surface types and airmass. Note that $\Theta_{sat}$ is the satellite zenith angle.

| Mask Subset | NN | | | JMA | | | BoM | | |
|---|---|---|---|---|---|---|---|---|---|
| Metric | KSS | TPR | FPR | KSS | TPR | FPR | KSS | TPR | FPR |
| $0° \leq \Theta_{sat} < 30°$ | 0.718 | 0.858 | 0.140 | 0.656 | 0.889 | 0.233 | 0.579 | 0.923 | 0.345 |
| $30° \leq \Theta_{sat} < 60°$ | 0.729 | 0.868 | 0.139 | 0.680 | 0.876 | 0.196 | 0.563 | 0.943 | 0.380 |
| $\Theta_{sat} \geq 60°$ | 0.658 | 0.853 | 0.195 | 0.499 | 0.690 | 0.191 | 0.376 | 0.934 | 0.558 |
| Land | 0.693 | 0.844 | 0.151 | 0.578 | 0.761 | 0.181 | 0.467 | 0.857 | 0.389 |
| Ocean | 0.685 | 0.863 | 0.178 | 0.587 | 0.794 | 0.207 | 0.451 | 0.956 | 0.505 |
| Day | 0.694 | 0.863 | 0.169 | 0.600 | 0.832 | 0.232 | 0.457 | 0.957 | 0.501 |
| Twilight | 0.661 | 0.888 | 0.226 | 0.458 | 0.627 | 0.170 | 0.320 | 0.904 | 0.584 |
| Night | 0.691 | 0.851 | 0.159 | 0.592 | 0.729 | 0.136 | 0.533 | 0.905 | 0.372 |
| $2 \leq$ Airmass $< 3$ | 0.658 | 0.838 | 0.180 | 0.532 | 0.766 | 0.234 | 0.404 | 0.935 | 0.531 |
| $3 \leq$ Airmass $< 4$ | 0.675 | 0.854 | 0.179 | 0.572 | 0.773 | 0.201 | 0.425 | 0.942 | 0.517 |
| Airmass $\geq 4$ | 0.682 | 0.861 | 0.179 | 0.572 | 0.799 | 0.227 | 0.462 | 0.936 | 0.475 |
| Overall | 0.691 | 0.859 | 0.168 | 0.589 | 0.787 | 0.198 | 0.472 | 0.936 | 0.464 |





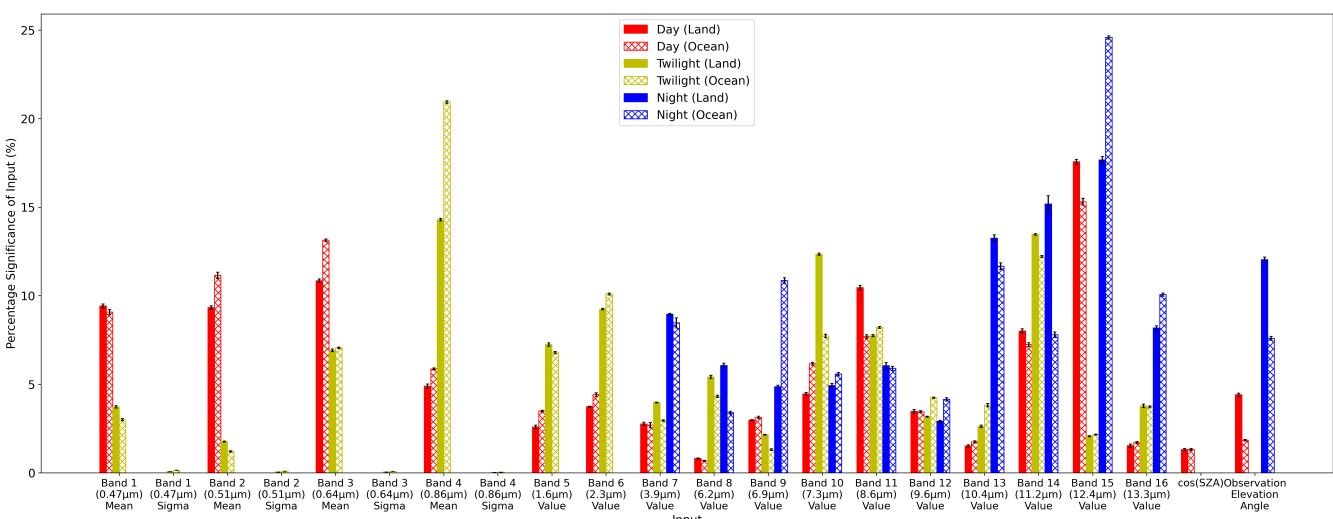

**Figure 6.** Plot summarising the output from the SHAP analysis. The SHAP values for each scenario, day, night, twilight, sea and land are normalised to 100. The results indicate the relative contribution of each input to defining the cloud mask.





**Figure 7.** A full-disk AHI scene for 01/11/2020 03:30 UTC. 3 RGB false colour composites of the scene are presented in the left-hand column, showing (a) true colour, (c) natural colour and (e) dust RGBs. The top, middle and bottom panels of the right-hand column show (b) the NN continuous cloud mask, (d) NN binary cloud mask and (f) a comparison of the 3 binary masks for the scene.







**Figure 8.** A partial disk AHI scene covering south-east Asia for 15/03/2021 03:30 UTC. 3 RGB false colour composites of the scene are presented in the left-hand column, showing (a) true colour, (c) natural colour and (e) dust RGBs. The top, middle and bottom panels of the right-hand column show (b) the NN continuous cloud mask, (d) NN binary cloud mask and (f) a comparison of the 3 binary masks for the scene.





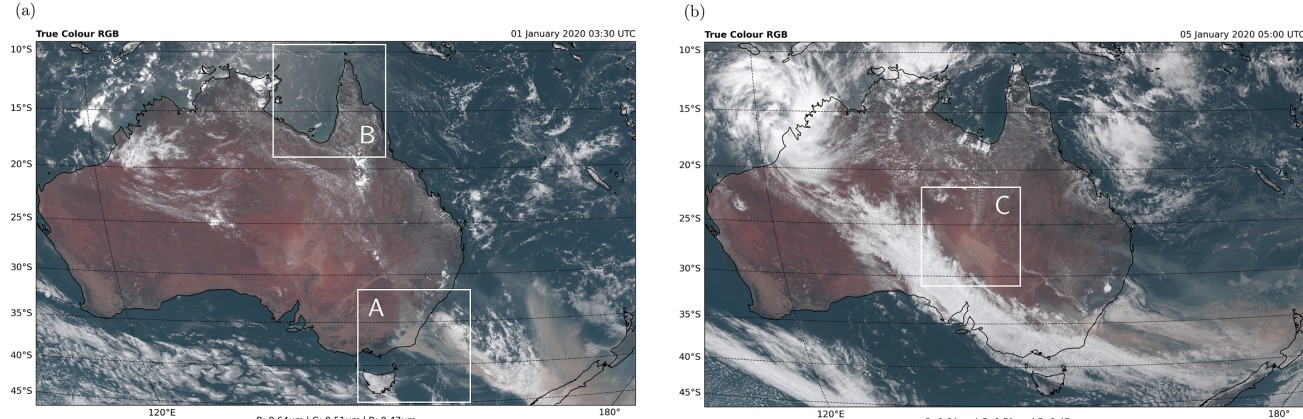

**Figure 9.** True colour RGB false colour composites from AHI scenes over Australia for (a) 01/01/2020 03:30 UTC and (b) 05/01/2020 05:00 UTC showing the approximate areas of 3 case-study regions. Region A contains an optically thick smoke plume on the SE coast of Australia, whilst region B contains a challenging scenario for cloud masks in the the Gulf of Carpentaria. Region C contains a thick dust plume in central Australia.



**Figure 10.** Region A on the SE coast of Australia from an AHI scene for 01/01/2020 03:30 UTC. 3 RGB false colour composites of the scene are presented in the left-hand column, showing (a) true colour, (c) natural colour and (e) dust RGBs. The top, middle and bottom panels of the right-hand column show (b) the NN continuous cloud mask, (d) NN binary cloud mask and (f) a comparison of the 3 binary masks for the scene.







**Figure 11.** Case study B covering the Gulf of Carpentaria from an AHI scene for 01/01/2020 03:30 UTC. 3 RGB false colour composites of the scene are presented in the left-hand column, showing (a) true colour, (c) natural colour and (e) dust RGBs. The top, middle and bottom panels of the right-hand column show (b) the NN continuous cloud mask, (d) NN binary cloud mask and (f) a comparison of the 3 binary masks for the scene.

**Figure 12.** Region C covering Central Australia and highlighting a dust storm for 01/01/2020 05:00 UTC. 3 RGB false colour composites of the scene are presented in the left-hand column, showing (a) true colour, (c) natural colour and (e) dust RGBs. The top, middle and bottom panels of the right-hand column show (b) the NN continuous cloud mask, (d) NN binary cloud mask and (f) a comparison of the 3 binary masks for the scene.