# Peer review of "Improving discrimination between clouds and optically thick aerosol plumes in geostationary satellite data"

_Atmospheric Measurement Techniques, 2021_

## Referee Comment (RC3)

**Review of "Improving discrimination between clouds and optically thick aerosol plumes in geostationary satellite data" by Robbins et al.**

Clouds are one of the prime remote sensing targets from space-based instruments which can provide critical information on the global distribution of clouds horizontally and vertically along with their phases. Clouds also present a serious problem for retrieval of aerosol and gas species from such instruments. Classifying a layer accurately as cloud is thus very important. In this paper, the authors describe the development and assessment of a cloud mask for Advanced Himawari Imager using a neural network algorithm and collocated CALIPSO data for training and validation. The paper is well within the ambit of AMT and all the sections are well organized and written clearly. In particular I enjoyed reading the section on case studies showing the strengths and caveats of the network. However I have some comments primarily dealing with the authors' use of CALIPSO data. I recommend publication of the paper after revision.

- 1. Since much of this work depends crucially on the Cloud Aerosol Discrimination (CAD) by CALIPSO, it is essential to be aware of some aspects of the latter. The authors state that they have analyzed both the 5km and 1km CALIPSO cloud products and then decided to go for the 1km product. Although the CAD algorithm in version 4 CALIPSO data has been extended to classify both the single shot (333m) and 1 km layers, the training sets used to generate the 5 dimensional pdfs that are used for the CAD algorithm never really used the optical properties of the 333m and 1km layers. In other words, the pdfs were generated using the measured variables of 5km layer products only and then applied to the 333m and 1km layers, so the quality of the CAD for 1km layers is largely unknown. Clearly this has ramifications for using the collocated CALIPSO dataset for training and validation. I would urge the authors to carry out a comparison of CAD at the two resolutions (5km vs 1km) to check on this. On line 102, the authors state that optimum results were found using the 1km layers. It is not clear how this was achieved, since no extinction retrievals are done for 1km cloud layers and an evaluation of the cloud optical depth bias being ingested in the training set is difficult to estimate. Perhaps a figure showing an example of the performance using both these resolutions and discussing the trade-off will be good.
- 2. I think that more stringent filtering of CALIPSO data will be needed for using in the training and validation of the neural networks. For instance, since late 2016, CALIOP has been having issues with low energy laser shots---these primarily affect the data quality above the South Atlantic Anomaly (SAA) region but are increasingly affecting other parts of the globe. This can lead to artifacts in the data including false layer detections at all altitudes, particularly in the dayside (see data advisory:

https://www-calipso.larc.nasa.gov/resources/calipso\_users\_guide/advisory/advisory\_2018-06-12/CALIPSO\_Laser\_Energy\_Technical\_Advisory.pdf ).

In fact one can identify some of these false layers as vertical streaks in Figure 4, say near 42°S. These false layer streaks probably have low CAD scores for the most part, which would take them out below 7.9 km by the CAD score filtering. It's not clear if the authors use the CAD score criterion above 7.9 km as well—if not, these false layer detections will

contaminate the training sets at higher altitudes. These effects can be alleviated by using a threshold of minimum laser energy field (say >0.08 joules) provided in 1 km/5km layer data files or vfm files.

- 3. Section 3.2 on CALIOP should be expanded giving some details of CALIPSO version 4 CAD algorithm and also more substantial references (see the special issue on CALIPSO version 4 algorithm in AMT).
- 4. Lines 165-170---what are the implications of this altitude threshold? The Labonne et al. (2009) paper that the authors use is rather old and more recent data from CALIPSO and other instruments indicate that biomass burning plumes can be injected at altitudes higher than 7.9 km. In particular the pyroCb events can transport smoke plumes to very high altitudes into the stratosphere. These high altitude smoke plumes often have high depolarization ratio similar to clouds and CALIPSO CAD algorithm classifies a good number of them as clouds which appear in CALIPSO browse images side by side with smoke layers as can be seen in the browse image below. This scene is from January 2, 2020 and the arrows mark smoke plumes downwind from the Black Summer bushfires similar to the one shown in Figure 10.

---

## Author Comment (AC1)

Thank you very much for your very kind review and comments. We have addressed the issues raised and amended the manuscript.

Please note, upon reflecting on some of the comments made about this study, we have rerun the models with labels set without the altitude limit to the CAD score filter. This has led to some minor improvements in the models' ability to classify cloud and aerosol when compared to the JMA and BoM masks. We have updated figures 4-8 and 10-12, as well as table 2, with the results from the new models (please see updated manuscript figures and tables section at the end of this document). With the exception of the case study of the dust storm over China, where the NN mask no longer misclassifies the dust plume, our analysis remains the same.

- In line 11, "0.106 and 0.198" and "0.314 and 0.464" have been replaced with "0.160 and 0.259" and "0.363 and 0.506" respectively.
- In line 12, "1.11 and 1.28 times" has been replaced with "1.13 and 1.29 times".
- We have removed the reference to the Labonne et al., 2009, study in line 196.
- In line 266, "a KSS of 0.691 versus 0.589 for the JMA product and 0.472 for the BoM product" has been replaced by "a KSS of 0.632 versus 0.523 for the JMA product and 0.432 for the BoM product".
- In lines 278-279, "the associated FPRs would be 0.314 versus 0.464 for NN and BoM algorithms and 0.106 versus 0.198 for the NN and JMA algorithms respectively" has been replaced with "the associated FPRs would be 0.363 versus 0.506 for NN and BoM algorithms and 0.160 versus 0.259 for the NN and JMA algorithms respectively".
- In line 279, "This implies that the NN accurately identifies 1.11 and 1.28 times" has been replaced with "This implies that the NN accurately identifies 1.13 and 1.29 times".
- In lines 336-340, "Over land, bands 4, 10 and 14 have approximately equivalent significance in the NN. Bands 4 and 14 serve the same role over land as they do over ocean. However, unlike over ocean, some land surface types can be bright in band 4 at twilight. This causes the NN to require a water vapour absorption band to effectively identify cloud over land during twilight and the NN has found band 10 to be most useful for this purpose" has been replaced with, "Over land, bands 11 and 14 have approximately equivalent significance in the NN. Bands 4 and 14 serve the same role over land as they do over ocean. However, unlike over ocean, some land surface types can be bright in band 4 at twilight. This causes the NN to require an additional cloud-detection band to effectively identify cloud over land during twilight and the NN has found band 11 to be most useful for this purpose".
- In lines 383-394, "However, all the masks fail to effectively classify the dust plume, with the exception of the NN mask accurately classifying a small section of the dust storm to the north of the Korean peninsula. Given that this event was a historically significant event with an unusually high plume (Filonchyk, 2022), the failure of the cloud masks might be expected. In particular, it shows that the NN cloud mask is only as effective as its training data and extreme events that it is not trained for will cause the mask to fail, although under more extreme scenarios than the JMA and BoM masks. In panel b of Fig. 8, pleasingly it can be seen that the section of the dust plume that is towards the centre of the scene is assigned scores significantly below values given to clouds - the plume has values of approximately 0.5, whereas clouds have values close to 1 - indicating that the NN mask is not confident the plume is cloud. A future algorithm could use this information within a convolutional NNs to improve the performance for large plumes or to develop uncertainty metrics" has been replaced with, "The JMA and BoM masks fail to effectively classify the dust plume, which the NN mask accurately identifies as non-cloud. Given that this event was a historically significant event with an unusually high plume (Filonchyk, 2022), the failure of the cloud masks might be expected. However, large areas of the dust plume are assigned relatively high values by the NN mask. In panel b of Fig. 8, it can be seen that the section of the dust plume that is towards the centre of the scene is assigned scores slightly below those assigned to cloud - the plume has values of approximately 0.5, whereas clouds have values close to 1 - indicating that, although the NN mask is not confident the plume is cloud, the dust storm poses a challenge to the NN masks classification algorithm. A future algorithm could use this information within convolutional NNs to improve the performance further for large plumes or to develop uncertainty metrics".

- In line 456, "the NN accurately detects 1.11 and 1.28 times" has been replaced with "the NN accurately detects 1.13 and 1.29 times".

**Comment 1:** l.152 The final inputs to the neural networks are listed. Line 91, however, states: "The auxiliary information from AHI is also included in the collocated data, such as the latitudes, longitudes, solar and observation angles." Are the latitudes and longitudes included as inputs for the neural networks?

**Reply 1:** Only the final inputs described in line 152 are used in the final models. All other auxiliary information was used for further analysis of the results. We have added a clarification of this point to line 184, which reads, "Only these inputs are used for the models. Auxiliary data, such as satellite zenith angle, latitude and longitude is used only for further analysis of results.".

**Comment 2:** l.172 The neural network training section lacks some minor details for reproducing the results. For example, how many epochs were used to train the neural networks? Was there an early stopping criterion to stop the optimization? How do the authors ensure the convergence of the trained neural networks?

**Reply 2:** The NNs are trained over 200 epochs. This number ensures convergence as the cost can be seen to asymptote before this value in all the models that were trained. Line 213 has been amended to include "Each NN is trained over 200 epochs to ensure convergence".

**Comment 3:** l.227 Please describe what is meant by a collocated dataset ("30 collocated datasets"). For example, do a single dataset correspond to some specific time instant, or are the pixels selected randomly, or something else?

**Reply 3:** We have added a short definition of a collocated dataset to line 256, which reads "where each dataset is a CALIOP overpass that has been collocated with AHI data".

**Comment 4:** l.252 Please clarify that by the surface type you mean land or ocean.

**Reply 4:** We have added the clarification of surface type to line 282, which reads "surface type (over ocean or over land)".

**Comment 5:** l.267 Typo "Shapely". It should be "Shapley".

**Reply 5:** The typo "Shapely" in line 296 has been corrected to "Shapley".

**Updated Manuscript Figures and Tables**

[Figure]

Figure 4 - Old

[Figure]

Figure 4 - Updated

[Figure]

Figure 5 - Old

[Figure]

Figure 5 - Updated

[Figure]

Figure 6 - Old

[Figure]

Figure 6 - Updated

[Figure]

Figure 7 - Old

[Figure]

Figure 7 - Updated

[Figure]

Figure 8 - Old

[Figure]

Figure 8 - Updated

[Figure]

Figure 10 - Old

Figure 10 - Updated

[Figure]

Figure 11 - Old

[Figure]

Figure 11 - Updated

[Figure]

Figure 12 - Old

Figure 12 - Updated

---

## Author Comment (AC2)

Thank you very much for your kind review and interesting comments. We have amended the manuscript to address the issues raised.

Please note, upon reflecting on some of the comments made about this study, we have rerun the models with labels set without the altitude limit to the CAD score filter. This has led to some minor improvements in the models' ability to classify cloud and aerosol when compared to the JMA and BoM masks. We have updated figures 4-8 and 10-12, as well as table 2, with the results from the new models (please see updated manuscript figures and tables section at the end of this document). With the exception of the case study of the dust storm over China, where the NN mask no longer misclassifies the dust plume, our analysis remains the same.

- In line 11, "0.106 and 0.198" and "0.314 and 0.464" have been replaced with "0.160 and 0.259" and "0.363 and 0.506" respectively.
- In line 12, "1.11 and 1.28 times" has been replaced with "1.13 and 1.29 times".
- We have removed the reference to the Labonne et al., 2009, study in line 196.
- In line 266, "a KSS of 0.691 versus 0.589 for the JMA product and 0.472 for the BoM product" has been replaced by "a KSS of 0.632 versus 0.523 for the JMA product and 0.432 for the BoM product".
- In lines 278-279, "the associated FPRs would be 0.314 versus 0.464 for NN and BoM algorithms and 0.106 versus 0.198 for the NN and JMA algorithms respectively" has been replaced with "the associated FPRs would be 0.363 versus 0.506 for NN and BoM algorithms and 0.160 versus 0.259 for the NN and JMA algorithms respectively".
- In line 279, "This implies that the NN accurately identifies 1.11 and 1.28 times" has been replaced with "This implies that the NN accurately identifies 1.13 and 1.29 times".
- In lines 336-340, "Over land, bands 4, 10 and 14 have approximately equivalent significance in the NN. Bands 4 and 14 serve the same role over land as they do over ocean. However, unlike over ocean, some land surface types can be bright in band 4 at twilight. This causes the NN to require a water vapour absorption band to effectively identify cloud over land during twilight and the NN has found band 10 to be most useful for this purpose" has been replaced with, "Over land, bands 11 and 14 have approximately equivalent significance in the NN. Bands 4 and 14 serve the same role over land as they do over ocean. However, unlike over ocean, some land surface types can be bright in band 4 at twilight. This causes the NN to require an additional cloud-detection band to effectively identify cloud over land during twilight and the NN has found band 11 to be most useful for this purpose".
- In lines 383-394, "However, all the masks fail to effectively classify the dust plume, with the exception of the NN mask accurately classifying a small section of the dust storm to the north of the Korean peninsula. Given that this event was a historically significant event with an unusually high plume (Filonchyk, 2022), the failure of the cloud masks might be expected. In particular, it shows that the NN cloud mask is only as effective as its training data and extreme events that it is not trained for will cause the mask to fail, although under more extreme scenarios than the JMA and BoM masks. In panel b of Fig. 8, pleasingly it can be seen that the section of the dust plume that is towards the centre of the scene is assigned scores significantly below values given to clouds - the plume has values of approximately 0.5, whereas clouds have values close to 1 - indicating that the NN mask is not confident the plume is cloud. A future algorithm could use this information within a convolutional NNs to improve the performance for large plumes or to develop uncertainty metrics" has been replaced with, "The JMA and BoM masks fail to effectively classify the dust plume, which the NN mask accurately identifies as non-cloud. Given that this event was a historically significant event with an unusually high plume (Filonchyk, 2022), the failure of the cloud masks might be expected. However, large areas of the dust plume are assigned relatively high values by the NN mask. In panel b of Fig. 8, it can be seen that the section of the dust plume that is towards the centre of the scene is assigned scores slightly below those assigned to cloud - the plume has values of approximately 0.5, whereas clouds have values close to 1 - indicating that, although the NN mask is not confident the plume is cloud, the dust storm poses a challenge to the NN masks classification algorithm. A future algorithm could use this information within convolutional NNs to improve the performance further for large plumes or to develop uncertainty metrics".

- In line 456, "the NN accurately detects 1.11 and 1.28 times" has been replaced with "the NN accurately detects 1.13 and 1.29 times".

**Comment 1:** Since much of this work depends crucially on the Cloud Aerosol Discrimination (CAD) by CALIPSO, it is essential to be aware of some aspects of the latter. The authors state that they have analyzed both the 5km and 1km CALIPSO cloud products and then decided to go for the 1km product. Although the CAD algorithm in version 4 CALIPSO data has been extended to classify both the single shot (333m) and 1 km layers, the training sets used to generate the 5 dimensional pdfs that are used for the CAD algorithm never really used the optical properties of the 333m and 1km layers. In other words, the pdfs were generated using the measured variables of 5km layer products only and then applied to the 333m and 1km layers, so the quality of the CAD for 1km layers is largely unknown. Clearly this has ramifications for using the collocated CALIPSO dataset for training and validation. I would urge the authors to carry out a comparison of CAD at the two resolutions (5km vs 1km) to check on this. On line 102, the authors state that optimum results were found using the 1km layers. It is not clear how this was achieved, since no extinction retrievals are done for 1km cloud layers and an evaluation of the cloud optical depth bias being ingested in the training set is difficult to estimate. Perhaps a figure showing an example of the performance using both these resolutions and discussing the trade-off will be good.

**Reply 1:** We have compared CAD scores between the 1km and 5km products, with 2 examples shown in figures 1 and 2 in the response specific figures section at the end of this document. These figures show the probability distribution of CAD scores for 2 overpasses that occurred during the 2019/2020 Black Summer bushfires at 1km and 5km resolution. One of these is from a night overpass, the other is from a day overpass. It can be seen in both cases that the general distribution of CAD scores is similar, although there are significantly fewer low CAD scores in the 1km product. Figure 3 shows a section of the CALIOP overpass starting at 1st January 2020 03:59:16 UTC. In this figure, we can see that the main reason for the difference between the 2 products comes from low CAD score layers; they are present in the 5km product, but are omitted in the 1km product. This is likely because they are misclassified thin aerosol layers that require increased horizontal sampling to determine. In terms of the impact on this study, we can see that the low CAD score layers are declared as non-cloud in the 1km product, and would be assigned the same label by our NNs. Therefore, we believe that the 1km product is suitable for this study.

- We have expanded section 3.4 to explain our reasoning, with lines 117-131 now reading as, "While the 5km algorithm is more sensitive to optically thin clouds, after initial investigation optimum results were found using the 1km L2 cloud-layer version 4.20 product (CAL\_LID\_L2\_01kmCLay-Standard-V4-20) (Young et al., 2018) because the higher spatial resolution leads to increased accuracy of identifying small-scale clouds in AHI pixels at 2km resolution. The version 4 product is used for this study due to improvements made in the cloud-aerosol discrimination (CAD) score algorithm for this product (Liu et al., 2019). The CAD algorithm seeks to discriminate between cloud and aerosol particles, such as fine spherical dust particles and water cloud, by using 5D probability density functions (PDF)s to assign values between -100 and 100 to each layer, with -100 being certainly aerosol and 100 being certainly cloud. The CAD algorithm improves on the previous version by applying the algorithm to all single shot retrievals, which were previously classified as cloud by default, as well as making improvements to identifying elevated aerosol layers and cloud layers under dense aerosols such as smoke (Liu et al., 2019). The version 4 algorithm is validated on the 5km product, but inspection of CAD scores between the 5km and 1km products indicate similar performance. Therefore, although the 5km product is more suitable for use with the CAD score, the 1km product is still appropriate for use in this study. However, it is important to note that extreme cases of aerosols can still lead to classification of aerosol layers as cloud from the CALIOP classification and that small scale (less then 1km across) clouds can be potentially misclassified and be a source of error in the NNs and validation, i.e. the pixel classification by CALIOP is assumed to be true throughout this study, but CALIOP misclassifying layers is a potential source of uncertainty within this study."

**Comment 2:** I think that more stringent filtering of CALIPSO data will be needed for using in the training and validation of the neural networks. For instance, since late 2016, CALIOP has been having issues with low energy laser shots---these primarily affect the data quality above the South Atlantic Anomaly (SAA) region but are increasingly affecting other parts of the globe. This can lead to artifacts in the data including false layer detections at all altitudes, particularly in the dayside (see data advisory: https://www-calipso.larc.nasa.gov/resources/calipso_users_guide/advisory/advisory_2018-06-12/CALIPSO_Laser_Energy_Technical_Advisory.pdf ). In fact one can identify some of these false layers as vertical streaks in Figure 4, say near 42oS. These false layer streaks probably have low CAD scores for the most part, which would take them out below 7.9 km by the CAD score filtering. It's not clear if the authors use the CAD score criterion above 7.9 km as well—if not, these false layer detections will contaminate the training sets at higher altitudes. These effects can be alleviated by using a threshold of minimum laser energy field (say >0.08 joules) provided in 1 km/5km layer data files or vfm files.

**Reply 2:** Unfortunately, we do not have the laser energy data included in our collocated data and re-running the entire collocation process is extremely computationally expensive and time-consuming. Future analysis will consider this information. However, we did not apply the CAD filter above 7.9km. We have since re-run the models with the CAD filter applied across all altitudes and updated all results accordingly. For more information, please see the statement given at the start of this document.

**Comment 3:** Section 3.2 on CALIOP should be expanded giving some details of CALIPSO version 4 CAD algorithm and also more substantial references (see the special issue on CALIPSO version 4 algorithm in AMT).

**Reply 3:** We have expanded the section regarding CALIOP by discussing the CAD algorithm in section 3.2 and including a fuller description of the caveats of using CALIOP as truth. At the end of section 3.2, lines  117-131 have been updated to include this information (please see reply 1).

**Comment 4:** Lines 165-170---what are the implications of this altitude threshold? The Labonne et al. (2009) paper that the authors use is rather old and more recent data from CALIPSO and other instruments indicate that biomass burning plumes can be injected at altitudes higher than 7.9 km. In particular the pyroCb events can transport smoke plumes to very high altitudes into the stratosphere. These high altitude smoke plumes often have high depolarization ratio similar to clouds and CALIPSO CAD algorithm classifies a good number of them as clouds which appear in CALIPSO browse images side by side with smoke layers as can be seen in the browse image below. This scene is from January 2, 2020 and the arrows mark smoke plumes downwind from the Black Summer bushfires similar to the one shown in Figure 10. (full CALIPSO imagery at: https://wwwcalipso.larc.nasa.gov/products/lidar/browse_images/show_v4_detail.php?s=production &v =V4-10&browse_date=2020-01-02&orbit_time=12-03- 45&page=3&granule_name=CAL_LID_L1-Standard-V4-10.2020-01-02T12-03- 45ZN.hdf ) Note that smoke layers at low altitudes are classified correctly as such by CALIPSO but at high altitudes near 12-15 km, there are a large number of layers which are likely misclassified as clouds. As the authors point out in section 5.3.3, there is possibly some ice mixed with smoke coming from the pyroCb event. However note the very high attenuated color ratios (right panel) towards the base of the layers. This is a tell-tale signature of smoke in CALIOP products and suggest that most of the layers should have been classified as smoke. This might relate to the cloud misclassification by NN in Figure 10 that the authors discuss in lines 376-384 and indicates the caveats in using CALIPSO as "truth label".

**Reply 4:** We agree with this point. Although layers with CAD scores less than 50 above 7.9km constitute appropriately just 2% of the total data, we re-ran the NNs with the altitude limit removed. The results show improved classification of the dust storm with minimal change to the the results of the other case studies and statistics. For more information, please see the statement at the start of this document.

**Comment 5:** Lines 227-233 and Figure 4. Please provide the CALIOP granule information and also state the month and day rather than in abbreviated form (for this Figure as well as others). Why is the CAD score scale going from -100 to 100—for clouds it should be only from 0-100. What are the blue layers poleward of 50oS and why are they appearing below the surface level?

**Reply 5:** Figure 4 has been updated with the CALIOP granule information and all figures have been updated with the month and day in full. The original plot for figure 4 is a standard diagnostic plot we used and includes the full range of CAD scores in case of using merged layer data. It has also been updated so that the range only goes from 0-100 and the surface artefacts that were plotted as blue have been removed for clarity.

**Comment 6:** The case study dealing with a strong dust storm (section 5.3.2) is somewhat intriguing. Asian dust and pollution events occur every year in spring (Huang et al., 2015, doi:10.1088/1748-9326/10/11/114018 , Di Pierro et al., 2011, doi:10.5194/acp-11-2225- 2011 ) and plumes from these events often travel at high altitudes ~ 3-8 km towards the arctic. One major problem with CALIPSO CAD in version 3 was the misclassification of these thick aerosol plumes as clouds, but these should have been corrected in version 4 through improved CAD. Therefore I am somewhat surprised that the NN didn't classify the 2021 dust storm correctly and once again wondering about the CALIOP CAD performance at 1km as possibly contributing to this.

**Reply 6:** Since we reran the NN model with the altitude limit removed, the dust plume over China is no longer misclassified (please see the statement at the start of this document for more information). Evidently, the relatively minor mislabelling due to this limit was causing the NN to incorrectly learn that elevated dust looks like cloud, but this has been rectified. In addition, we believe this indicates that the 1km product is not at fault.

**Comment 7:**  Typo in line 413: "bask"→ "mask"

**Reply 7:** Thank you, "bask" has been corrected to "mask" in line 453.

**Comment 8:** Line 94: CALIPSO→ CALIOP and Calipso→ CALIPSO

**Reply 8:** This has been corrected. "CALIPSO" and "Calipso" have been corrected to "CALIOP" and "CALIPSO" respectively in line 110.

**Comment 9:** Line 339 and Figure 7: I had a hard time finding pale blue color in the sun glint area in the mask comparison plot!

**Reply 9:** An improved description of the region and its location have been added to line 367, which reads, "At approximately 10◦ S, towards the centre of the scene, there is a small region of sunglint. Some very small areas within this region, concentrated just north of Australia, are shown as pale blue in the mask comparison, indicating that the JMA and BoM cloud masks agree on the classification of this region but the NN mask disagrees with the other masks.".

**Comment 10:** Line 90: it would be better to specify the input variables for which the mean and standard deviation are being used.

**Reply 10:** Lines 107-108 has been amended to read, "The mean and the standard deviation of these high resolution channels (channels 1-4) over the downsampled area are calculated and considered as inputs into the NN.".

**Comment 11:** In Figures 7,8 and 10-12, in the cloud mask comparison panels, dark blue is labelled as "all clear"—should it be "non-cloud" instead---as stated on line 328.

**Reply 11:** This is correct, panel f in figures 7, 8 and 10-12 have been updated to read "All non-cloud" rather than "all clear".

**Response Specific Figures**

[Figure]

Figure 1: Probability density distribution for 1st January 2020 03:59:16 UTC overpass during the day.

[Figure]

Figure 2: Probability density distribution for 1st January 2020 14:42:52 UTC overpass during the night.

[Figure]

Figure 3: Curtain plots of a section of a CALIOP overpass from 1st January 2020 03:59:16 UTC overpass during the day at 1km resolution (bottom) and 5km resolution (top) that passes over Australia.

**Updated Manuscript Figures and Tables**

[Figure]

Figure 4 - Old

[Figure]

Figure 4 - Updated

[Figure]

Figure 5 - Old

[Figure]

Figure 5 - Updated

[Figure]

Figure 6 - Old

[Figure]

Figure 6 - Updated

[Figure]

Figure 7 - Old

[Figure]

Figure 7 - Updated

[Figure]

Figure 8 - Old

[Figure]

Figure 8 - Updated

[Figure]

Figure 10 - Old

Figure 10 - Updated

[Figure]

Figure 11 - Old

[Figure]

Figure 11 - Updated

[Figure]

Figure 12 - Old

Figure 12 - Updated

---

## Author Comment (AC3)

Thank you for your review and comments. We have amended the manuscript to address the issues raised.

Please note, upon reflecting on some of the comments made about this study, we have rerun the models with labels set without the altitude limit to the CAD score filter. This has led to some minor improvements in the models' ability to classify cloud and aerosol when compared to the JMA and BoM masks. We have updated figures 4-8 and 10-12, as well as table 2, with the results from the new models (please see updated manuscript figures and tables section at the end of this document). With the exception of the case study of the dust storm over China, where the NN mask no longer misclassifies the dust plume, our analysis remains the same.

- In line 11, "0.106 and 0.198" and "0.314 and 0.464" have been replaced with "0.160 and 0.259" and "0.363 and 0.506" respectively.
- In line 12, "1.11 and 1.28 times" has been replaced with "1.13 and 1.29 times".
- We have removed the reference to the Labonne et al., 2009, study in line 196.
- In line 266, "a KSS of 0.691 versus 0.589 for the JMA product and 0.472 for the BoM product" has been replaced by "a KSS of 0.632 versus 0.523 for the JMA product and 0.432 for the BoM product".
- In lines 278-279, "the associated FPRs would be 0.314 versus 0.464 for NN and BoM algorithms and 0.106 versus 0.198 for the NN and JMA algorithms respectively" has been replaced with "the associated FPRs would be 0.363 versus 0.506 for NN and BoM algorithms and 0.160 versus 0.259 for the NN and JMA algorithms respectively".
- In line 279, "This implies that the NN accurately identifies 1.11 and 1.28 times" has been replaced with "This implies that the NN accurately identifies 1.13 and 1.29 times".
- In lines 336-340, "Over land, bands 4, 10 and 14 have approximately equivalent significance in the NN. Bands 4 and 14 serve the same role over land as they do over ocean. However, unlike over ocean, some land surface types can be bright in band 4 at twilight. This causes the NN to require a water vapour absorption band to effectively identify cloud over land during twilight and the NN has found band 10 to be most useful for this purpose" has been replaced with, "Over land, bands 11 and 14 have approximately equivalent significance in the NN. Bands 4 and 14 serve the same role over land as they do over ocean. However, unlike over ocean, some land surface types can be bright in band 4 at twilight. This causes the NN to require an additional cloud-detection band to effectively identify cloud over land during twilight and the NN has found band 11 to be most useful for this purpose".
- In lines 383-394, "However, all the masks fail to effectively classify the dust plume, with the exception of the NN mask accurately classifying a small section of the dust storm to the north of the Korean peninsula. Given that this event was a historically significant event with an unusually high plume (Filonchyk, 2022), the failure of the cloud masks might be expected. In particular, it shows that the NN cloud mask is only as effective as its training data and extreme events that it is not trained for will cause the mask to fail, although under more extreme scenarios than the JMA and BoM masks. In panel b of Fig. 8, pleasingly it can be seen that the section of the dust plume that is towards the centre of the scene is assigned scores significantly below values given to clouds - the plume has values of approximately 0.5, whereas clouds have values close to 1 - indicating that the NN mask is not confident the plume is cloud. A future algorithm could use this information within a convolutional NNs to improve the performance for large plumes or to develop uncertainty metrics" has been replaced with, "The JMA and BoM masks fail to effectively classify the dust plume, which the NN mask accurately identifies as non-cloud. Given that this event was a historically significant event with an unusually high plume (Filonchyk, 2022), the failure of the cloud masks might be expected. However, large areas of the dust plume are assigned relatively high values by the NN mask. In panel b of Fig. 8, it can be seen that the section of the dust plume that is towards the centre of the scene is assigned scores slightly below those assigned to cloud - the plume has values of approximately 0.5, whereas clouds have values close to 1 - indicating that, although the NN mask is not confident the plume is cloud, the dust storm poses a challenge to the NN masks classification algorithm. A future algorithm could use this information within convolutional NNs to improve the performance further for large plumes or to develop uncertainty metrics".

- In line 456, "the NN accurately detects 1.11 and 1.28 times" has been replaced with "the NN accurately detects 1.13 and 1.29 times".

**Comment 1:** One of the issues is the innovative contribution of this paper. As the author mentioned, using Machine Learning to facilitate satellite image recognition/categorization is a hot topic. Many studies have tried using ML/CNN to identify cloud and/or aerosols from passive sensors, for example Marais et al., 2020, Lee et al., 2021, Wang et al., 2020. Some of these studies also uses lidar as benchmark to label particle types. The new contribution from this study that is differ from the already published studies shall be clarified.

**Reply 1:** These studies have been added to the manuscript and a short description of the way in which this study differs from those cited has also been added to the manuscript.
- Marais et al., 2020, has been cited alongside Hughes et al., 2019, in line 72 critiquing hand-labeled datasets.
- Wang et al., 2020, and Lee et al., 2021, have been cited in line 80 and the way in which this work differs described in lines 81-86, which read, "These studies all demonstrate the success of machine learning algorithms trained on data labeled by active instruments, but do not extend the technique to geostationary instruments, which require a more sophisticated collocation technique to account for the parallax between the active and passive sensors. These studies do not focus on biomass burning plumes, which are of particular significance over Australia. In addition, in this paper we combine this technique with explainable machine learning models to better understand the influence of passive instrument channels on the outcome of the classification.".

**Comment 2:** In addition, more information of the disadvantage and advantage of passive and active remote sensing techniques of clouds and aerosols are needed to justify the benefits of using active sensor to provide typing information.

**Reply 2:** Further discussion of the advantages and disadvantages of passive and active sensors have been added to the manuscript to reinforce our justification for using labels assigned by active sensors.
- A critique of passive instruments has been added to lines 38-40, which reads, "However, individual passive sensors can only see in 2D. For example, to classify whether a bright, cold pixel is snow/ice or a cloud top, a retrieval algorithm must be applied to the pixel to classify it. These algorithms require evaluation using ground-based instruments and active instruments to ensure that they are accurate".
- A discussion about the advantages of using active sensors has been added to lines 42-45, which reads, "As they have their radiation source on-board, active instruments can operate independently of solar illumination and are more sensitive to thin atmospheric layers, such as thin cirrus and aerosols, than passive instruments. In addition, active sensors can retrieve the height of layers within a pixel by evaluating the strength of the return signal and time taken for the pulse to return. This makes them able to detect clouds and aerosols within their pixels much more accurately than passive instruments.".
- Finally, we have added a summary of the advantages of using active sensors for labeling in lines 47-50, which reads, "Combining the temporal and spatial resolution of passive instruments with the more accurate classification of atmospheric layers achieved by active sensors is desirable to create an optimal algorithm for classifying clouds and aerosols. By using active instruments to label passive sensor pixels, classification algorithms for passive sensors can be developed that take advantage of the increased accuracy of active sensors.".

**Comment 3:** Discussions on potential misclassification in CALIOP of identify spherical fine particles as clouds and how that is going to impact the outcome of this study needs to be discussed in the article. Related to this issue, my biggest concern is that there is little information of the uncertainty/QA procedures used when using CALIOP CAD to identify aerosols and clouds. The CAD > 50 thresholds will likely mark some of the small clouds as aerosols, which is shown in Figure 12. The upper right corner has many fine popcorn clouds, which is marked as potential cloudy in ML output and identified as clear in binary mask. In contrast, the other two cloud products marked this area as cloudy. This can

cause large problem in aerosol retrieval. Due to this mislabeling is caused by how clouds are defined, it will not be marked as missing detection of clouds in validation (accuracy score). Plus, an altitude threshold of CAD will mark some elevated aerosols as clouds, such as volcano eruption/stratosphere aerosols, although the percentage of these data will be very small.

**Reply 3:** This point has been addressed along with comments made by reviewer 3 by discussing the CAD algorithm in section 3.2 and including a more complete description of the caveats of using CALIOP as truth.

- At the end of section 3.2, lines 117-131 now read, "While the 5km algorithm is more sensitive to optically thin clouds, after initial investigation optimum results were found using the 1km L2 cloud-layer version 4.20 product (CAL\_LID\_L2\_01kmCLay-Standard-V4-20) (Young et al., 2018) because the higher spatial resolution leads to increased accuracy of identifying small-scale clouds in AHI pixels at 2km resolution. The version 4 product is used for this study due to improvements made in the cloud-aerosol discrimination (CAD) score algorithm for this product (Liu et al., 2019). The CAD algorithm seeks to discriminate between cloud and aerosol particles, such as fine spherical dust particles and water cloud, by using 5D probability density functions (PDF)s to assign values between -100 and 100 to each layer, with -100 being certainly aerosol and 100 being certainly cloud. The CAD algorithm improves on the previous version by applying the algorithm to all single shot retrievals, which were previously classified as cloud by default, as well as making improvements to identifying elevated aerosol layers and cloud layers under dense aerosols such as smoke (Liu et al., 2019). The version 4 algorithm is validated on the 5km product, but inspection of CAD scores between the 5km and 1km products indicate similar performance. Therefore, although the 5km product is more suitable for use with the CAD score, the 1km product is still appropriate for use in this study. However, it is important to note that extreme cases of aerosols can still lead to classification of aerosol layers as cloud from the CALIOP classification and that small scale (less then 1km across) clouds can be potentially misclassified and be a source of error in the NNs and validation, i.e. the pixel classification by CALIOP is assumed to be true throughout this study, but CALIOP misclassifying layers is a potential source of uncertainty within this study.".

Using the CAD score is, in of itself, a QA procedure that has been used in other studies (Winker et al., 2013, (https://doi.org/10.5194/acp-13-3345-2013); Watson-Parris et al., 2018 (https://doi.org/10.1002/2013JD019527)) and is used in the same way as it has been in our study. However, we have acknowledged that the CAD score may lead to some small clouds being misclassified due to the CAD score.

With regards to the altitude limit, please see the statement given at the start of this document.

**Comment 4:** It is also not clear to me how the NN model is set up. Is small batch of horizontal pixel from AHI used as input. If so, what is the size of batch? How is CALIOP labeling work for each batch?

**Reply 4:** The model uses information from a single pixel at 2km resolution and uses this to classify the pixel. This is done for every pixel in an AHI scene. The phrase "to analyse a scene pixel-by-pixel" has been added to line 179 to clarify this point.

**Comment 5:** In terms of validation, due to the ambiguity in determine CALIOP cloud and thick aerosols, external data, such as ground lidar can be used to validate the cloud/aerosol mask as well as more cases of intense smoke from wildfire and pollution are needed.

**Reply 5:** This is a great idea. Unfortunately, we could not identify any good quality LIDAR data for the case studies over Australia. However, there are proposals to use weather radar for tracking biomass burning plumes, although this is still in the very early stages of development and may be available for future studies.

**Comment 6:** Another suggestion is that if the main purpose of the model is to separate thick aerosols from clouds while maintain reliable cloud mask, instead of comparing the cloud/aerosol mask to

other cloud mask products, comparisons between ML cloud mask to cloud mask within other aerosol products is more appropriate. Because cloud mask, which is made to remove "unclear" sky, is known to have "clear sky bias"; while aerosol products try their best to preserve these aerosol scenes.

**Reply 6:** This is an interesting idea. However, from visual inspection of aerosol product masks, it can be seen that these masks also have issues with removing cloud (see MAIAC mask for MODIS taken during the 2019/2020 bushfires in response-specific figures at the end of this document). For example, the JAXA AOD product is based on the deep-blue method (She et al., 2020 (http://dx.doi.org/10.3390/rs12244125)) over ocean and operates in a similar fashion to the JAXA standard cloud mask. It is not obvious that comparing against these masks will make a difference to the statistical analysis, but including these in future studies may be considered.

**Comment 7:** For reader's benefits, reword the description of the parallax correction. From my understand, the pseudo-CALIOP vertical profile is generated using layer information from different CALIOP lidar pulse along the AHI airmass pathway. However, the description of the parallax correction is very confusing mentioning the angle from CALIOP needs to match angles from AHI.

**Reply 7:** This description has been reworded to clarify that the AHI observation angle for each layer is what must be matched to account for parallax.
- Lines 154-161 now read, "The parallax correction for each layer was performed by:
  – Calculating the observation angles of the CALIOP layer as it would be seen by AHI at the position and altitude specified in the CALIOP data, i.e. the angle that corresponds to the dashed line beneath the cloud layer in Fig. 2.
  – The observation angles of the CALIOP layer as seen by AHI were then matched with the observation angles for AHI corresponding to the Earth's surface.
  – Where the AHI observation angles matched, the layer was assigned to the collocated AHI pixel, i.e. the cloud layer in Fig. 2 would be assigned to the pixel that corresponds with the red star. As the match is to the closest pixel, this leads to a spatial uncertainty of approximately ±1km at nadir for AHI.
  – This was repeated for every layer and a pseudo-CALIOP profile was generated for each AHI pixel. This includes thin layers that AHI may struggle to observe and are accepted as a potential source of error in the final cloud mask."

**Response-specific figures**

[Figure]

The MAIAC product from 1st January 2020, showing the retrieval mask misclassifying cloud.

**Updated Manuscript Figures and Tables**

[Figure]

Figure 4 - Old

[Figure]

Figure 4 - Updated

[Figure]

Figure 5 - Old

[Figure]

Figure 5 - Updated

[Figure]

Figure 6 - Old

[Figure]

Figure 6 - Updated

[Figure]

Figure 7 - Old

[Figure]

Figure 7 - Updated

[Figure]

Figure 8 - Old

[Figure]

Figure 8 - Updated

[Figure]

Figure 10 - Old

Figure 10 - Updated

[Figure]

Figure 11 - Old

[Figure]

Figure 11 - Updated

[Figure]

Figure 12 - Old

Figure 12 - Updated